# Gut butyrate-producers confer post-infarction cardiac protection

Hung-Chih Chen[1], Yen-Wen Liu[2], Kuan-Cheng Chang[3,4], Yen-Wen Wu [5], Yi-Ming Chen[1], Yu-Kai Chao [1], Min-Yi You[1], David J. Lundy [6], Chen-Ju Lin[1], Marvin L. Hsieh[7], Yu-Che Cheng[1], Ray P. Prajnamitra[1], Po-Ju Lin[1], Shu-Chian Ruan[1], David Hsin-Kuang Chen [1], Edward S. C. Shih[1], Ke-Wei Chen[3], Shih-Sheng Chang[3,4], Cindy M. C. Chang[7], Riley Puntney[8], Amy Wu Moy [8], Yuan-Yuan Cheng[1], Hsin-Yuan Chien[1], Jia-Jung Lee[9], Deng-Chyang Wu[10], Ming-Jing Hwang [1], Jennifer Coonen[8], Timothy A. Hacker[7], C-L. Eric Yen [1,11], Federico E. Rey [12], Timothy J. Kamp [13] & Patrick C. H. Hsieh [1,13,14] ✉

The gut microbiome and its metabolites are increasingly implicated in several cardiovascular diseases, but their role in human myocardial infarction (MI) injury responses have yet to be established. To address this, we examined stool samples from 77 ST-elevation MI (STEMI) patients using 16 S V3-V4 next-generation sequencing, metagenomics and machine learning. Our analysis identified an enriched population of butyrate-producing bacteria. These findings were then validated using a controlled ischemia/reperfusion model using eight nonhuman primates. To elucidate mechanisms, we inoculated gnotobiotic mice with these bacteria and found that they can produce beta-hydroxybutyrate, supporting cardiac function post-MI. This was further confirmed using HMGCS2-deficient mice which lack endogenous ketogenesis and have poor outcomes after MI. Inoculation increased plasma ketone levels and provided significant improvements in cardiac function post-MI. Together, this demonstrates a previously unknown role of gut butyrate-producers in the post-MI response.

The human microbiome, comprising more than 10 trillion bacteria, plays important roles in homeostasis and disease progression in multiple organ systems[1–5]. These interactions can be mediated, directly or indirectly, by bacteria-derived metabolites. For example,

gut microbial production of trimethylamine (TMA), and its hepatic metabolism into trimethylamine-N-oxide (TMAO), increase the risk of cardiovascular diseases and myocardial infarction (MI)[6,7]. Phenylacetylglutamine, produced by gut microbial phenylalanine

[1]Institute of Biomedical Sciences, Academia Sinica, Taipei 115, Taiwan. [2]Division of Cardiology, Department of Internal Medicine, National Cheng Kung University Hospital, College of Medicine, National Cheng Kung University, Tainan 704, Taiwan. [3]Division of Cardiovascular Medicine, China Medical University Hospital, Taichung 40447, Taiwan. [4]School of Medicine, China Medical University, Taichung 40402, Taiwan. [5]Cardiovascular Medical Center, Far Eastern Memorial Hospital, New Taipei City 220, Taiwan. [6]Graduate Institute of Biomedical Materials and Tissue Engineering, Taipei Medical University, Taipei 110, Taiwan. [7]Model Organisms Research Core, Department of Medicine, University of Wisconsin-Madison, Madison, WI 53705, USA. [8]Wisconsin National Primate Research Center, University of Wisconsin–Madison, Madison, WI 53715, USA. [9]Division of Nephrology, Department of Medicine, Kaohsiung Medical University & Hospital, Kaohsiung 807, Taiwan. [10]Division of Gastroenterology, Department of Medicine, Kaohsiung Medical University & Hospital, Kaohsiung 807, Taiwan. [11]Department of Nutritional Sciences, University of Wisconsin-Madison, Madison, WI 53706, USA. [12]Department of Bacteriology, University of Wisconsin-Madison, Madison, WI 53706, USA. [13]Department of Medicine and Stem Cell and Regenerative Medicine Center, University of Wisconsin-Madison, Madison, WI 53705, USA. [14]Institute of Medical Genomics and Proteomics and Institute of Clinical Medicine, National Taiwan University College of Medicine, Taipei 100, Taiwan. ✉e-mail: phsieh@ibms.sinica.edu.tw

metabolism, triggers platelet activation through adrenergic receptors, leading to increased thrombosis and MI[8]. On the other hand, we have previously shown, in mice, that the gut microbiome-derived short chain fatty acid (SCFA) propionate modulates the host immune system and plays a protective role in MI[9]. However, these animal studies do not necessarily translate to human patients due to large differences in microbiome composition.

In humans, studies of coronary artery disease (CAD) have revealed the importance of the microbiome and its metabolites. Depletion of *Acetobacter* of the Proteobacteria phylum and enrichment of *Betaproteobacteria* of the Burkholderiales order were associated with dysmetabolism in CAD[10]. Comparing CAD and control cohorts in the context of type 2 diabetes revealed the pathophysiological relevance of gut microbiome-associated dysmetabolism in the prodromal phase of CAD. A similar multiomic profiling of a CAD cohort identified significant depletion of SGB4712, a previously unknown bacterial species of the *Clostridiaceae* family[11]. This SGB4712 species was linked to decreased phenylacetylglutamine and increased ergothioneine metabolites, suggesting the cardioprotective potential of this species. These studies provide insights into the role of gut microbiota in the development of cardiovascular diseases. However, the roles of the gut microbiome and metabolites during acute post-injury cardiac repair in humans remains to be explored. Identifying novel aspects of the injury response may provide new options for therapies which are urgently needed in the clinic.

Therefore, in the present study we aimed to examine the timeline of gut microbiota and plasma metabolite changes that occur following acute ST-elevation myocardial infarction (STEMI), compared to a control cohort. We recruited STEMI patients and applied integrated metagenomic approaches to address the clinical significance of the alterations. To validate the key results from our human study, we then used nonhuman primates, which have a more similar microbiome and MI injury response to humans. Eight rhesus macaques were subjected to cardiac ischemia/reperfusion (IR) injury under highly-controlled conditions. Lastly, germ-free (GF), specific pathogen free (SPF) and HMGCS2-deficient mice were treated with key bacterial strains or metabolites to reveal the underlying mechanisms of cardioprotection. Together, these findings provide significant new knowledge regarding the gut-heart-axis and can aid in the development of potential prediction and therapeutic targets for STEMI.

## Results

### Butyrate-producing gut bacteria are enriched following STEMI

To investigate the alteration of the gut microbiome in STEMI, we performed 16S V3-V4 next-generation sequencing (NGS) on a total of 214 stool samples from n = 77 STEMI patients (confirmed by a board-certified cardiologist using electrocardiograph and angiography) and n = 70 age and BMI-matched controls (Ctrl) (Fig. 1a and Supplementary Table 1). Fecal samples from STEMI patients were collected at two time points; within three days of percutaneous coronary intervention (PCI) (STEMIT1) and again approximately 28 days after PCI (STEMIT2). STEMI and Ctrl samples differed significantly in multivariate analysis. The STEMIT1 group had distinct amplicon sequence variants (ASVs) in comparison with the Ctrl and STEMIT2 groups. More than 70% of the bacteria were shared by both sexes and patients with/without hyperlipidemia (Fig. 1b and Supplementary Figs. 1a and 2a). Higher α-diversity was observed in the STEMIT1 group compared with the STEMIT2 group (Shannon's index, accounting for both abundance and evenness of species), regardless of sex and hyperlipidemia (Fig. 1c and Supplementary Figs. 1b and 2b). Compared with the Ctrl group, both α-diversity and β-diversity (both weighted and unweighted UniFrac distance) were higher in the STEMIT1 group, representing a more diverse microbiota in this group (Fig. 1c, d and Supplementary Fig. 3a). An increase in *Firmicutes/Bacteroidetes* ratio has previously been reported

as a marker for hypertension[12]. Similarly, we observed a two-fold increase in *Firmicutes/Bacteroidetes* ratio in the STEMIT1 group compared with both the Ctrl and STEMIT2 groups (Supplementary Fig. 3b). However, there was no significant difference of the Firmicutes/Bacteroidetes ratio between normal and hypertensive groups in our samples (Supplementary Fig. 2d). The left ventricular (LV) ejection fraction (EF) of STEMI patients was inversely correlated with the Shannon's index and Pielou's evenness (Fig. 1e). To further identify the affected gut microbes, we performed Linear discriminant analysis (LDA) Effect Size (LEfSe) analysis at the genus level with the 16S V3-V4 NGS results[13]. Whereas commensal bacteria *Bifidobacterium* were reduced in STEMIT1 samples, *Desulfovibrio* and *Lactobacillus* were more abundant in the STEMIT1 and STEMIT2 samples, respectively[14] (Fig. 1f and Supplementary Figs. 1c and 2c). Notably, *Anaerotruncus*, *Alistipes*, *Butyricimonas*, *Enterococcus*, *Holdemanella* and *Subdoligranulum*, some of whose members have been reported to produce butyrate[15,16], were enriched in STEMIT1 samples (Fig. 1f and Supplementary Figs. 1c, 2c, and 3c). Analyzing over time, the gut microbial communities showed indistinguishable composition and diversity at the acute phase of STEMI and revealed clear enrichment of *Anaerotruncus*, *Alistipes*, *Butyricimonas*, *Enterococcus*, *Holdemanella* and *Subdoligranulum* at D3-4 (Supplementary Fig. 3d–g). A deep scale metagenome shotgun analysis of sixteen representative samples (five Ctrl and eleven STEMIT1 samples) extended LEfSe to the species level, showing reduction of *Bifidobacterium alolescentis* and *Bifidobacterium ruminantium*, and expansion of *Butyricimonas virosa*, *Streptococcus parasanguinis* and *Streptococcus salivarius* in STEMIT1 samples (Fig. 1g). Changes in these bacteria were further validated with species-specific primer sets using quantitative PCR (Fig. 1h and Supplementary Table 2). Of note, this is different to our previous study in mice which found alterations in *Lactobacillus*[9]. This is likely due to significant differences between baseline murine and human microbiome composition; across 70 human subjects, *Lactobacillus* represented only $0.25 \pm 0.06\%$ of the identified bacteria, compared to $26.25 \pm 5.61\%$ in mice.

### Nonhuman primate ischemia/reperfusion model recapitulates enrichment of gut butyrate-producers

The human-derived data above have limitations, including baseline differences between the control and STEMI groups, such as hyperlipidemia. We also lacked pre-MI stool samples, and could not control for diet, socioeconomic status or other variables which influence the microbiome[17]. Samples were also obtained during a time range of 1–3 days post-MI. Therefore, we carried out a controlled experiment using nonhuman primates which share significant genetic, physiological biochemical and metabolic similarities with humans[18]. Rhesus macaques (n = 8) were subjected to cardiac ischemia for 90 min followed by reperfusion (IR model). The duration of artery occlusion, reperfusion time and collection of stool samples (pre-IR −1D, post-IRD1, D7 and D28) were precisely controlled. We then profiled changes in the gut microbiota (Fig. 1i). The gut microbiota compositions were distinct at each time point (Fig. 1j). Moreover, both α- and β-diversities increased after IR and peaked at IRD28 and IRD7, respectively (Fig. 1k, l and Supplementary Fig. 4a). Similar to the human STEMIT1 samples, the macaque *Firmicutes/Bacteroidetes* ratio was upregulated at IRD1 (Supplementary Fig. 4b). *Butyricimonas*, *Faecalibacterium*, *Holdemanella*, *Roseburia* and *Subdoligranulum* were enriched after injury (Fig. 1m). We also confirmed reduction of *Bifidobacterium alolescentis*, and enrichment of *Butyricimonas virosa*, *Streptococcus parasanguinis*, *Streptococcus salivarius* and *Subdoligranulum variabile* after IR using qPCR (Fig. 1n and Supplementary Table 2), following the same trend we observed in humans. These conserved increases in butyrate-producing bacteria after cardiac injury in both human and nonhuman primates indicate that they may influence post-injury cardiac repair.

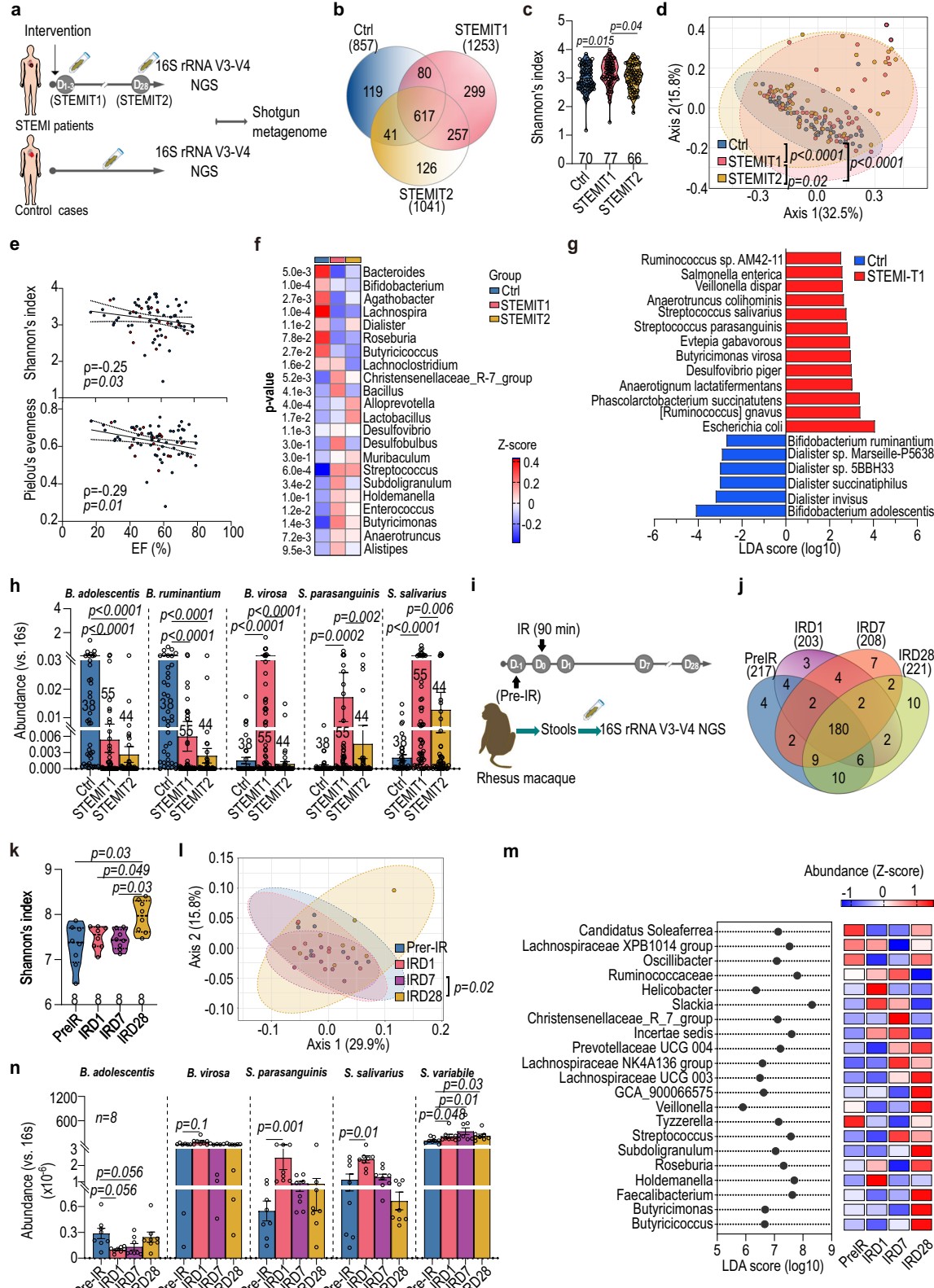

## Machine learning strategy for gut microbiome-driven STEMI diagnosis

Precision diagnosis of CVDs requires a combination of physical examination and specialized testing such as echocardiography, catheterization and blood tests. Previous reports suggest that alteration of gut microbiota can serve as biomarkers for diseases such as colorectal cancer, inflammatory bowel disease and CVD[19,20]. To investigate the possibility of using gut microbial composition as a diagnostic tool for STEMI, we performed supervised machine learning on the features of bacterial taxa using the PyCaret package to classify STEMI and control samples (Fig. 2a). By means of random data splitting, 70% of the samples were used as a training set to build the prediction model, while the remaining 30% were assigned as a test set to evaluate the model performance (Fig. 2a). We first trained the

**Fig. 1 | Enrichment of butyrate-producing gut microbiome after cardiac injury.** **a** Stools from ST-elevation myocardial infarction (STEMI) patients were collected right after percutaneous intervention (PCI) (STEMIT1) and on D28 after PCI (STE-MIT2). The controls (Ctrl) and STEMI stools were subjected to **b**–**f** 16S rRNA V3-V4 NGS and **g-h** metagenome shotgun. **b** Venn diagram showing overlapping amplicon sequence variants (ASVs) of Ctrl, STEMIT1 and STEMIT2. **c** Shannon's index of Ctrl and STEMI gut microbiota (vs. STEMIT1). **d** Principal co-ordinates analysis (PCoA) of weighted Unifrac of Ctrl and STEMI gut microbiota (vs. STEMIT1). **e** Spearman correlation of cardiac ejection fraction (EF, %) with Shannon's index (upper) and Pielou's evenness (lower). Blue for non-diabetic; red for diabetic. **f** Differentially abundant genera in Ctrl and STEMI samples. **g** Linear discriminant analysis (LDA) scores computed with distinct species in Ctrl and STEMIT1 via metagenomic ana-lysis of 5 Ctrl and 11 STEMIT1 samples. **h** qPCR confirmation of *Bifidobacterium adolescentis* (*B. adolescentis*), *Bifidobacterium ruminantium* (*B. ruminantium*), *Butyricimonas virosa* (*B. virosa*), *Streptococcus parasanguinis* (*S. parasanguinis*) and

*Streptoocccus salivarius* (*S. salivarius*) abundance (vs. Ctrl for *B. adolescentis*, *B. ruminantium*; vs. STEMIT1 for *B. virosa*, *S. parasanguinis*, *S. salivarius*). **i** Nonhuman primates (NHPs) were subjected to cardiac ischemic/reperfusion (IR) injury. The stools at pre-IR, IRD1, IRD7 and IRD28 were subjected to 16S rRNA V3-V4 NGS. **j** Venn diagram showing overlapping ASVs in NHP stools at IR, IRD1, IRD7 and IRD28. **k** Shannon's index of NHP gut microbiota in response to IR (vs. IRD28). **l** PCoA of weighted Unifrac of NHP gut microbiota (vs. IRD28). **m** LDA scores (left) and abundance (right) computed with distinct features in NHP gut microbiota in response to IR. **n** qPCR confirmation of *B. adolescentis*, *B. virosa*, *S. parasanguinis*, *S. salivarius* and *Subdoligranulum variabile* (*S. variabile*) abundance in NHP stools (vs. pre-IR). The number of biologically independent animals are indicated in each chart. Data in **c, f, h, k, n** were analyzed with Kruskal–Wallis test followed by Dunn's correction; data in **d, i** were analyzed with PERMANOVA. Data are represented as mean ± SEM.

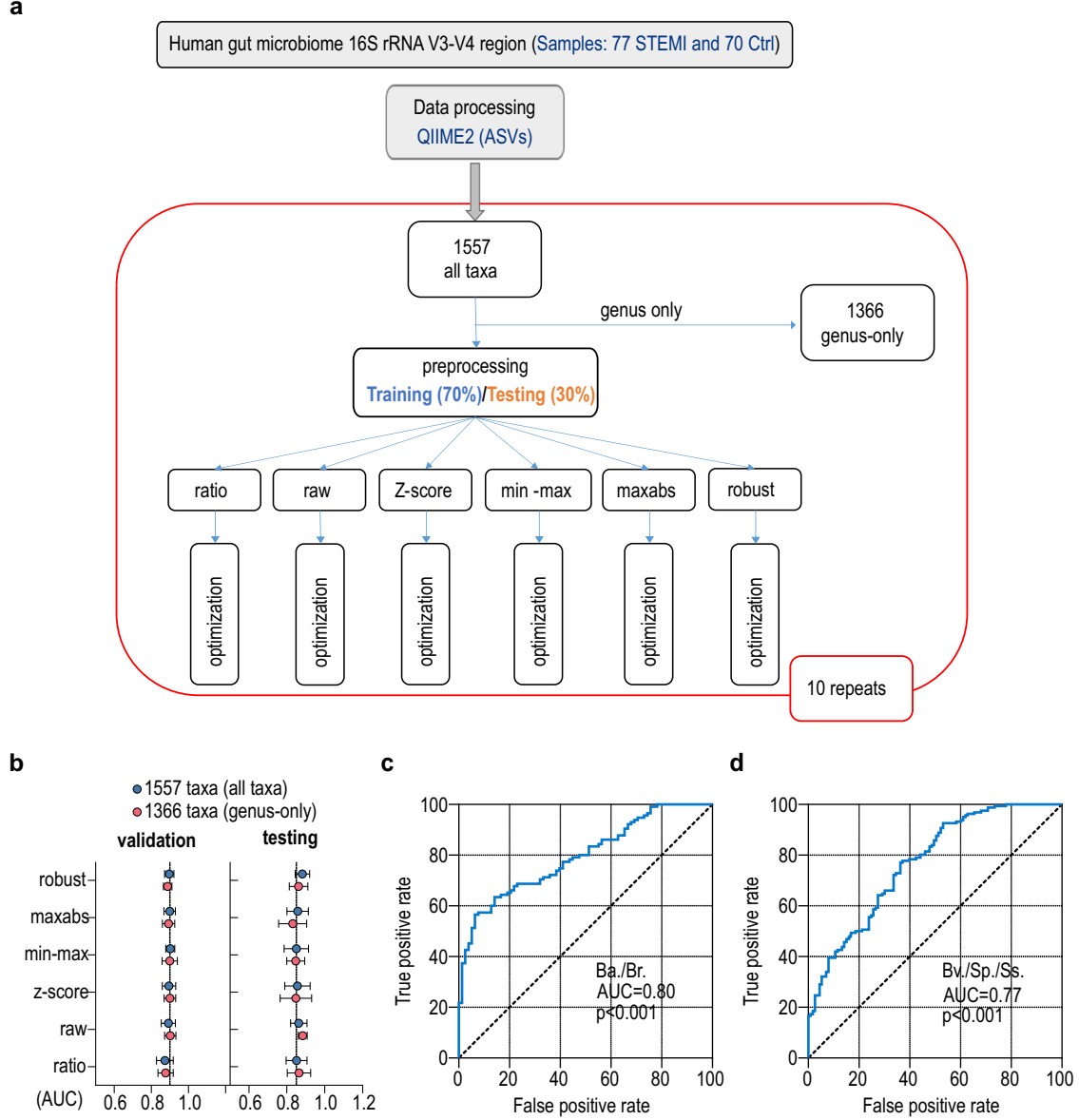

**Fig. 2 | Microbiome-driven precision diagnostics for STEMI using machine learning.** **a** Workflow for supervised machine learning. **b** The performance (recorded as area under curve, AUC) of microbiota-based model with different algorithms for the diagnosis of myocardial infarction. *n* = 10-fold cross-validations. **c** Receiver operating characteristic (ROC) curve performance using Ctrl-

predominant bacteria *Bifidobacterium adolescentis* (*Ba.*) and *Bifidobacterium ruminantium* (*Br.*). **d** ROC curve performance using STEMI-predominant bacteria include *Streptococcus parasanguinis* (*Sp.*), *Streptococcus salivarius* (*Ss.*) and *Butyr-icimonas virosa* (*Bv.*). ROC curves were analyzed with Wilson/Brown test with 95% confidence interval.

prediction models with either all (1557 taxonomic features at all levels) or genus-only (1366 taxonomic features at the genus level) taxa under different normalization, including ratio, raw, z-score, min-max scaling, maximum absolute scaling (maxabs) and robust scaling (Fig. 2b and Supplementary Data 1). During testing, both all-taxa and genus-only taxa analysis showed comparable area under curve (AUC) of more than 0.85 with different algorithms (Fig. 2b and Supplementary Table 3), indicating the potential for accurate determination of STEMI patients using gut microbiota configuration. Comparable AUCs were obtained after model calibration (Supplementary Table 4). This performance is equivalent to, or better than, reported machine learning studies in predicting diseases using gut microbiome sequencing data[19,20] (Supplementary Table 5). To further evaluate the diagnostic value of the control-predominant *B. alolescentis*/*B. ruminantium* and STEMI-predominant *B. virosa*/*S. parasanguinis*/*S. salivarius* identified by shotgun metagenome, we performed the receiver operating characteristic (ROC) curve analysis using the abundance distribution of these bacteria in Ctrl and STEMI groups by qPCR experiment. The ROC curve showed AUC of 0.80 and 0.77 for control-predominant and STEMI-predominant bacteria respectively, suggesting the favorable power of these bacteria for STEMI diagnosis (Fig. 2c, d).

## STEMI fecal microbiome transplantation deteriorates post-injury cardiac function in germ-free mice

The gut microbial composition has a profound impact on the host health. In the STEMIT1 samples, we observed an enrichment of both TMA-associated bacteria and butyrate-producing bacteria (Fig. 1f). To experimentally assess the influence of the human gut microbiota on post-injury cardiac function in a controlled manner, we transplanted human fecal samples from Ctrl and STEMIT1 into 12-week-old male C57Bl/6 J germ-free (GF) mice[21]. Interestingly, we noted a modest decrease in baseline cardiac function of mice receiving STEMI-FMT (Fig. 3a, b and Supplementary Fig. 5a). After challenging the mice with MI surgery via left descending coronary artery ligation, there was unexpectedly high mortality in the mice which received STEMIT1 fecal microbiome transplantation (FMT) (Fig. 3b). This suggests that the STEMI-derived gut microbiota caused an worsened response to MI. Thus, to further investigate the effects of STEMI microbiota on post-injury cardiac function, we utilized a milder injury model using angiotensin II (1.44 mg/kg/day) for fourteen days (Fig. 3c, d and Supplementary Fig. 5b). Principle co-ordinates analysis discriminated gut microbiomes from Ctrl- and STEMI-FMT mice (Fig. 3e). Additionally, the Firmicutes/Bacteroidetes ratio was higher in the gut microbiota of STEMI-FMT mice, and the ratio was also elevated in Ctrl-FMT mice treated with angiotensin II (Fig. 3f). Compared with Ctrl-FMT, the STEMI-FMT mice showed worsened LV EF and fraction shortening (FS) as well as reduced LV end diastolic and systolic volume (EDV and ESV) (Fig. 3g and Supplementary Fig. 5c). Likewise, STEMI-FMT increased LV stiffness and compromised contractility of recipient mice, showing higher EDPVR as well as lower ESPVR, PRSW and dP/dt max (vs. EDV) (Supplementary Fig. 5d). This could be attributed to cardiomyocyte hypertrophy (Fig. 3h). In addition, STEMI-FMT mice had thinner submucosa and shorter villi in the intestine (Fig. 3i), suggestive of increased gut inflammation[22]. We also found increased circulating proinflammatory cytokines such as IL-17 and IL-22 even before angiotensin II challenge in STEMI-FMT mice (Fig. 3j and Supplementary Fig. 6).

## Ketone body metabolism was enriched during post-injury cardiac repair

Bacterially-produced small molecules are a key driver of the microbiome activity[4,23]. Shotgun metagenomics analysis of the Ctrl and STEMIT1 stools revealed STEMIT1-associated enrichment of bacterial genes involved in the metabolism of amino acids, short chain fatty acids and the TCA cycle (Fig. 4a). An increase in bacterial genes involved in butyrate metabolism coincided with accumulation of

butyrate-producing bacteria in STEMIT1 samples (Figs. 1g and 4a). Of note, bacterial genes for ketone body metabolism were also augmented in the STEMIT1 samples (Fig. 4a). To determine the changes of plasma metabolites in STEMI samples, we performed both NMR and LC-MS metabolomics (Fig. 4b). Initial large-scale NMR metabolomics screening revealed distinct clustering of Ctrl, STEMIT1 and STEMIT2 metabolites (Fig. 4c). ROC curves based on the enrichment of metabolites in ketone body metabolism showed AUCs of about 0.901, suggesting alteration of ketone body metabolism as a potential prediction marker for STEMI (Fig. 4d). This agrees with a previous report showing the correlation of circulating β-hydroxybutyrate and the outcome of STEMI[24]. In addition, we also observed increased levels of plasma Trimethylamine-N-oxide (TMAO) in STEMIT1 samples, corresponding to the enrichment of TAM-associated bacteria in STEMIT1 stool samples (Figs. 1f and 4e). Metabolic pathway analysis from LC-MS metabolomics showed enrichment of fatty acid, amino acid and ketone body metabolism as well as biosynthesis of polyamines in STEMIT1 plasma compared with both Ctrl and STEMIT2 plasma, highlighting drastic metabolic alterations in the acute phase of MI (Fig. 4f). Compared with Ctrl, metabolism of butyrate, glutamate and pyruvate, and biosynthesis of bile acid, carnitine and spermidine/spermine were persistently enriched in the plasma of both STEMIT1 and STEMIT2, suggesting that these were STEMI-associated metabolic changes (Fig. 4f). Increased β-hydroxybutyrate in STEMI plasma was further validated with a colorimetric assay (Fig. 4g). We next addressed whether similar metabolism alteration could be observed in the better-controlled nonhuman primate IR model (Fig. 4h). Similar to the human samples, metabolic alteration began early in the acute phase after IR, showing enrichment of nearly all metabolic pathways examined at IRD1 compared with pre-IR and IRD28, such as butyrate, pyruvate and glutamate metabolism (Fig. 4i). Colorimetric assay confirmed that plasma β-hydroxybutyrate increased at IRD1 and then reduced over time (Fig. 4j). Together these data highlight the importance of butyrate and ketone body metabolism after cardiac injury.

## Butyrate supplementation confers better post-MI cardiac function preservation under intact gut microbiota

From metabogenomic analyses, we identified an increase in butyrate-producing bacteria after cardiac injury in both humans and nonhuman primates (Figs. 1 and 4). Previous studies have found therapeutic effects of butyrate following MI[25,26]. However, this was not investigated within the context of the microbiome. Therefore, we used a combination of broad-spectrum antibiotics (ABX) to deplete the C57BL/6J mouse host microbiome, then supplemented mice with butyrate via gavage beginning one day after MI and continuing for twenty days (Fig. 5a). Supplementation successfully increased the levels of fecal butyrate and plasma β-hydroxybutyrate (Fig. 5b, c). Interestingly, plasma levels of β-hydroxybutyrate were higher when the host gut microbiota was intact, suggesting a possible contribution of bacterial ketogenesis (Fig. 5c and Supplementary Fig. 7). In addition, enrichment of amino acid metabolism and glycolysis induced by butyrate supplementation was higher when the host gut microbiota was intact (Supplementary Fig. 7a, c). Improved EF and FS were noticed in the butyrate-supplemented groups, agreeing with previous studies. However, this was especially so in mice with an intact gut microbiota (Fig. 5d, e and Supplementary Fig. 8). Supplementing mice with butyrate reduced the post-MI infarct size (Fig. 5f, g). Cardiac mechanical properties were also improved, particularly in the context of intact gut microbiota, presenting higher ESPVR, PRSW and dP/dt max (vs. EDV) as well as lower EDPVR (Fig. 5h). Moreover, the butyrate treatment showed a dose-dependent cardioprotection (Supplementary Fig. 9). Together these data illustrate the importance of intact gut commensal microbiota for the protective role of butyrate during post-MI cardiac repair.

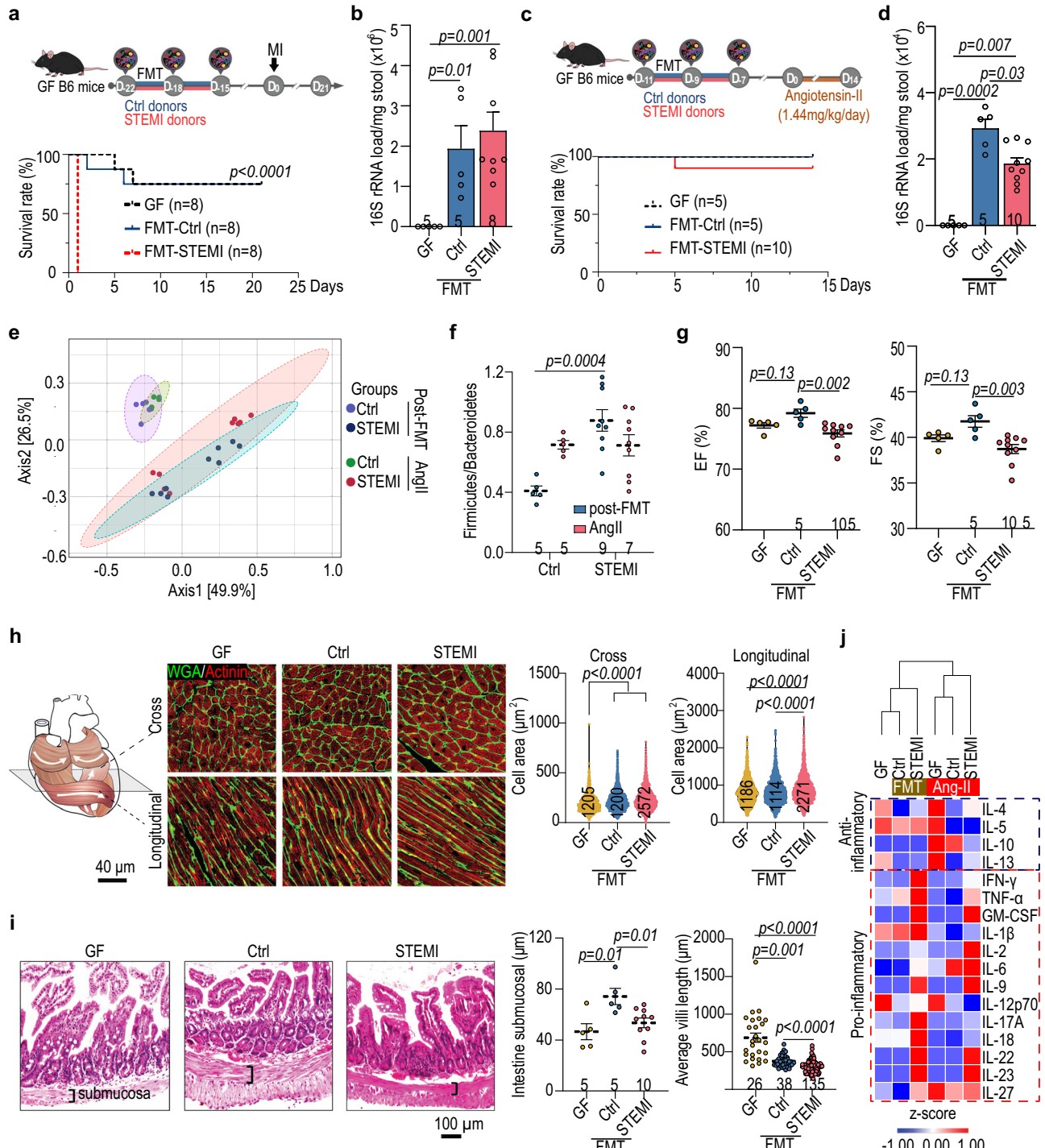

**Fig. 3 | STEMI fecal microbiota transplantation in germ-free mice deteriorates post-injury cardiac repair.** Twelve-week-old germ-free (GF) mice were colonized with human fecal microbiota for 7 days before cardiac injury. **a** Experimental design for myocardial infarction (MI) model on fecal microbiota transplantation (FMT)-GF mice (upper panel). The survival rate of FMT-GF mice subjected to MI for 21 days (lower panel). Data were analyzed with log rank (Mantel−Cox) test with 95% confidence interval. **b** qPCR-based determination of fecal bacterial load for the FMT-GF mice subjected to MI (vs. Ctrl). **c** Experimental design for angiotensin II (AngII) challenge on FMT-GF mice (upper panel). The survival curve of FMT-GF mice subjected to AngII challenge for 14 days (lower panel). **d** The fecal bacterial load for the FMT-GF mice administrated with AngII. **e** Differential grouping of the gut microbiota in Ctrl and STEMI samples in unweighted principal co-ordinates analysis (PCoA). **f** The ratio of gut Firmicutes relative to Bacteroidetes in FMT mice after cardiac injury. **g** Echocardiographic analysis of left ventricular ejection fraction (EF,

%) and fraction shortening (FS, %) in FMT-GF mice on day 14 after AngII challenge. **h** Changes in the size of cardiomyocytes in FMT mice subjected to AngII challenge for 14 days. The illustration of muscle orientation in the heart (upper left) and the representative immunofluorescent staining of cardiac tissues with WGA-488 co-stained with actinin (upper right). The statistical analysis of cardiomyocyte size is shown in the lower panel with cell number listed in the inset of each bar. **i** The colonic pathophysiology of GF mice receiving control and STEMI-FMT were examined with hematoxylin and eosin staining. The statistical analysis of sub-mucosal thickness and villi length of proximal colon is shown the right panel. **j** The changes of anti-inflammatory and pro-inflammatory cytokines in GF mice receiving control and STEM FMT, determined using multiplex immunoassays. The number of biologically independent mice are indicated in each chart. Kruskal−Wallis test followed by FDR correction was used to analyze data in (**b, d, g**−**i**) two-way ANOVA was used to analyze data in (**f**). Data are represented as mean ± SEM.

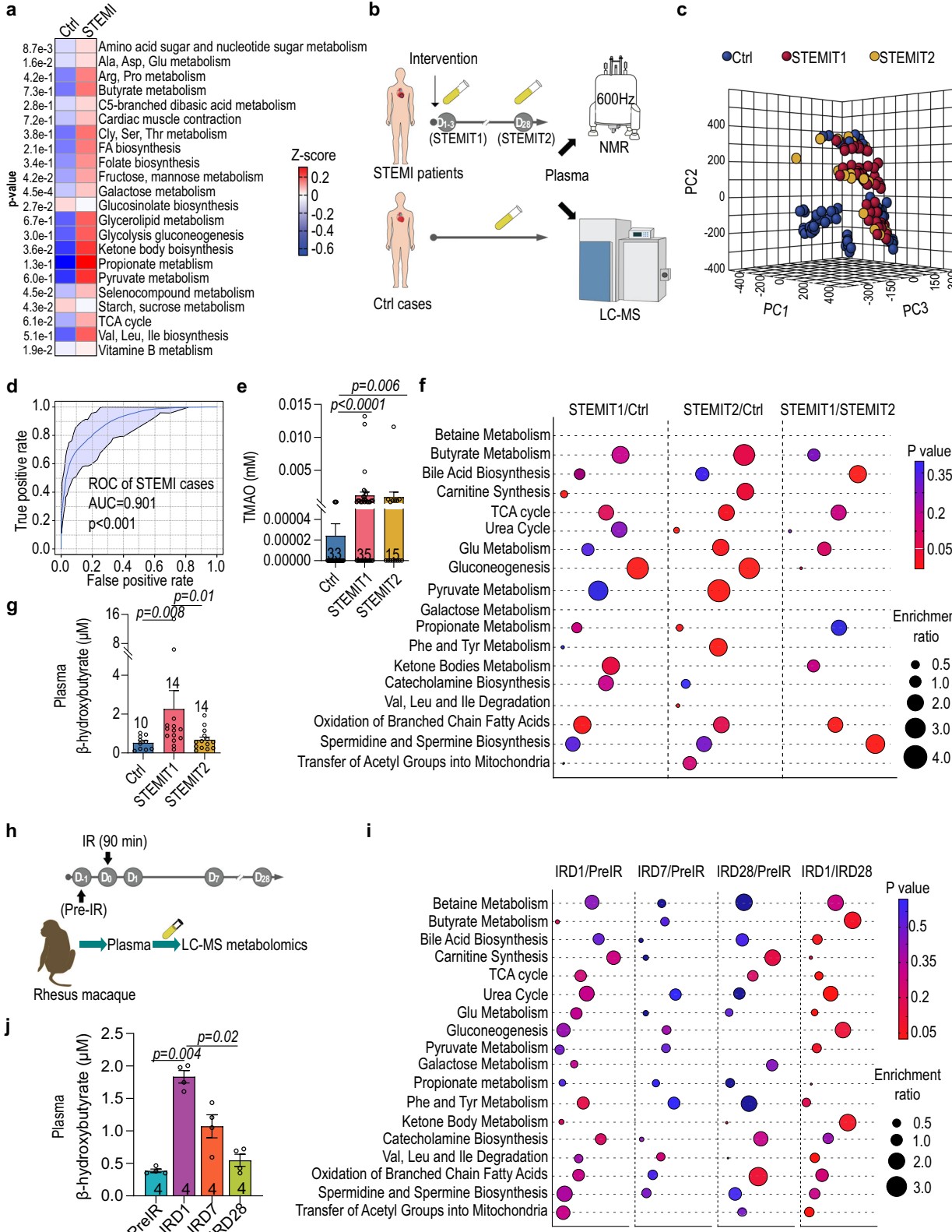

### Colonization of butyrate-producing bacteria ameliorates post-MI cardiac injury

The shotgun metagenomics showed that bacterial genes encoding key-step enzymes for ketone body metabolism (*ACAT*, *HMGCS*, *OXCT* and *BDH*) were enriched in the STEMI samples and the bacteria of note included STEMIT1-associated *B. adolescentis*, *B. virosa* and *S. parasanguinis* (Fig. 6a). *B. adolescentis*, *B. virosa* and *S. parasanguinis* were

all capable of producing butyrate in culture, although that *B. adolescentis* was less potent (Fig. 6b). Interestingly, all three bacteria appeared to produce β-hydroxybutyrate, according to colorimetric assay (Fig. 6c) which we then confirmed by LC-MS (Supplementary Fig. 10a and Supplementary Table 6). While residual β-hydroxybutyrate was detected in plain culture media, these data provide a possible explanation for the additional rise in plasma β-

**Fig. 4 | Ketone body biosynthesis and degradation pathways are enriched in STEMI patients. a** Enrichment of metabolic pathways revealed by shotgun metagenomics with Ctrl and STEMIT1 samples. **b** Schematic illustration of human plasma metabolomics profiling using nuclear magnetic resonance (NMR) and liquid chromatography–mass spectrometry (LC-MS). **c** Partial Least Squares Discriminant Analysis (PLS-DA) of Ctrl, STEMIT1 and STEMIT2 based on NMR plasma metabolite profiling. **d** ROC curve for STEMI prediction based on the enrichment of lactate, pyruvate, acetone, acetoacetate, glutamate, 3-hydroxybutyrate and butyrate based on NMR plasma metabolite profiling. **e** The plasma level of Trimethylamine-N-oxide (TMAO) in human control and STEMI samples determined with NMR (vs. Ctrl). **f** Metabolic pathway enrichment in the Ctrl, STEMIT1 and STEMIT2 plasma samples via LC-MS analysis. The enrichment ratio was computed by observed hits/expected

hits. **g** The level of human plasma β-hydroxybutyrate using colorimetric assay (vs. STEMIT1). **h** Nonhuman primates subjected to cardiac ischemic/reperfusion (IR) injury for ninety minutes and the plasma samples were collected at −1D (pre-IR), D1 (IRD1), D7 (IRD7) and D28 (IRD28) and subjected to LC-MS metabolomics profiling. **i** Enrichment of the metabolic pathways of rhesus macaque at pre-IR, IRD1, IRD7 and IRD28 using LC-MS. **j** The level of rhesus macaque plasma β-hydroxybutyrate using colorimetric assay (vs. IRD1). The number of biologically independent samples are indicated each chart. Data in **a, f, i** were analyzed with two-sided unpaired Student's *t*-test, data **e, g, j** were analyzed with Kruskal–Wallis test followed by FDR correction. ROC curves were analyzed with Wilson/Brown test with 95% confidence interval. Data are represented as mean ± SEM.

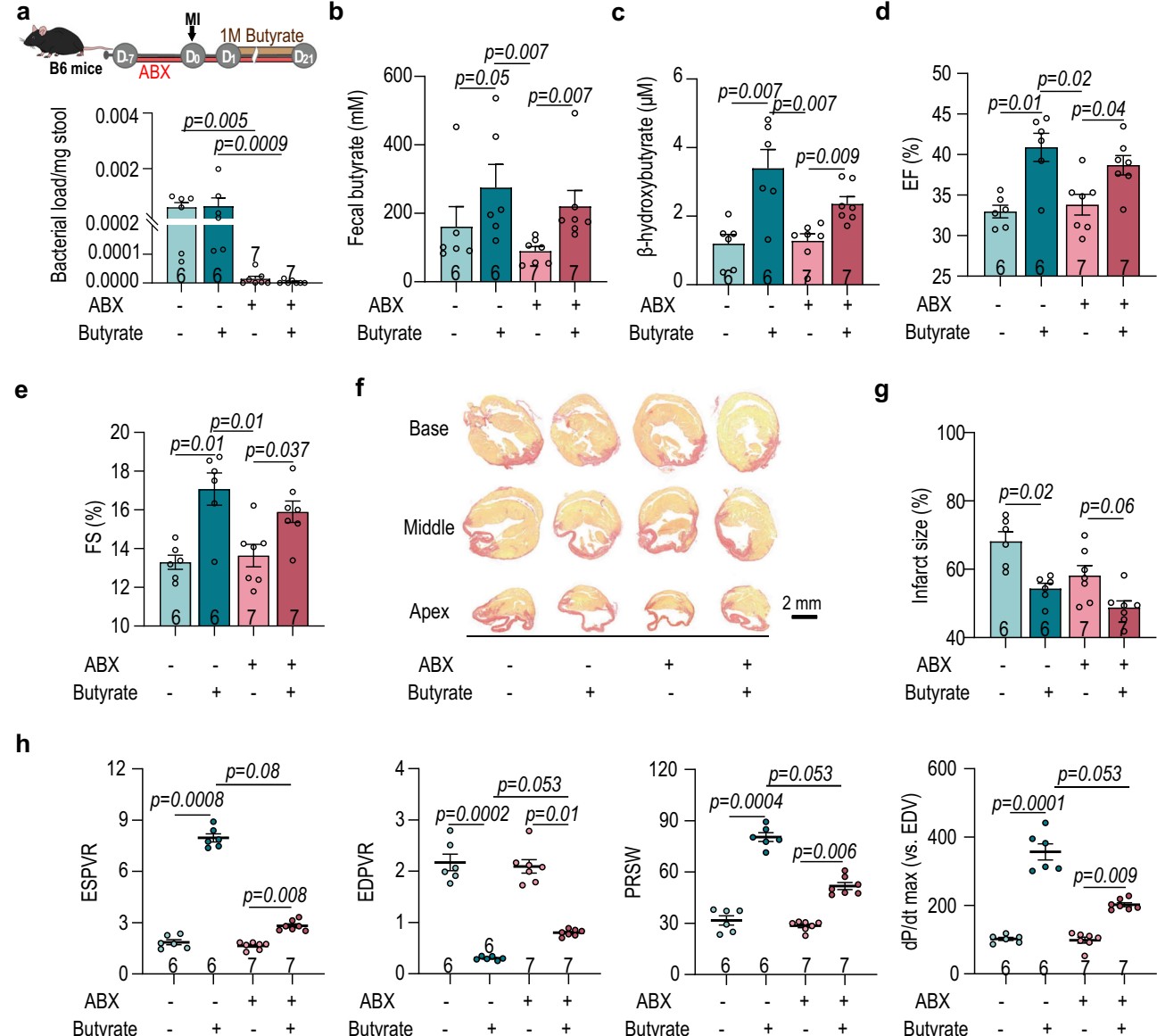

**Fig. 5 | Butyrate supplementation confers better preservation of cardiac function after myocardial injury. a** Experimental design for the effects of butyrate on post-MI cardiac repair (upper panel) in SPF mice. Half of the mice were also treated with antibiotics (ABX) to deplete the gut microbiota. The relative bacterial load in feces on MI day 21 was determined with 16S rRNA qPCR (lower panel). **b** Level of fecal butyrate on MI day 21 using HPLC analysis. **c** Plasma level of β-hydroxybutyrate on MI day 21 using colorimetric assay. **d–e** Changes in left

ventricular **d** EF and **e** FS in response to butyrate supplementation on MI day 21. Survival rate of SPF mice 21-days post-MI. **f** Representative images of cardiac infarct size labeled with picrosirus red and **g** quantification. **h** Catheterization analysis of post-MI cardiac function, including ESPVR, EDPVR, PRSW and dP/dt max (vs. EDV). The number of biologically independent mice are indicated in each chart. Data were analyzed with Kruskal–Wallis test followed by FDR correction. Data are represented as mean ± SEM.

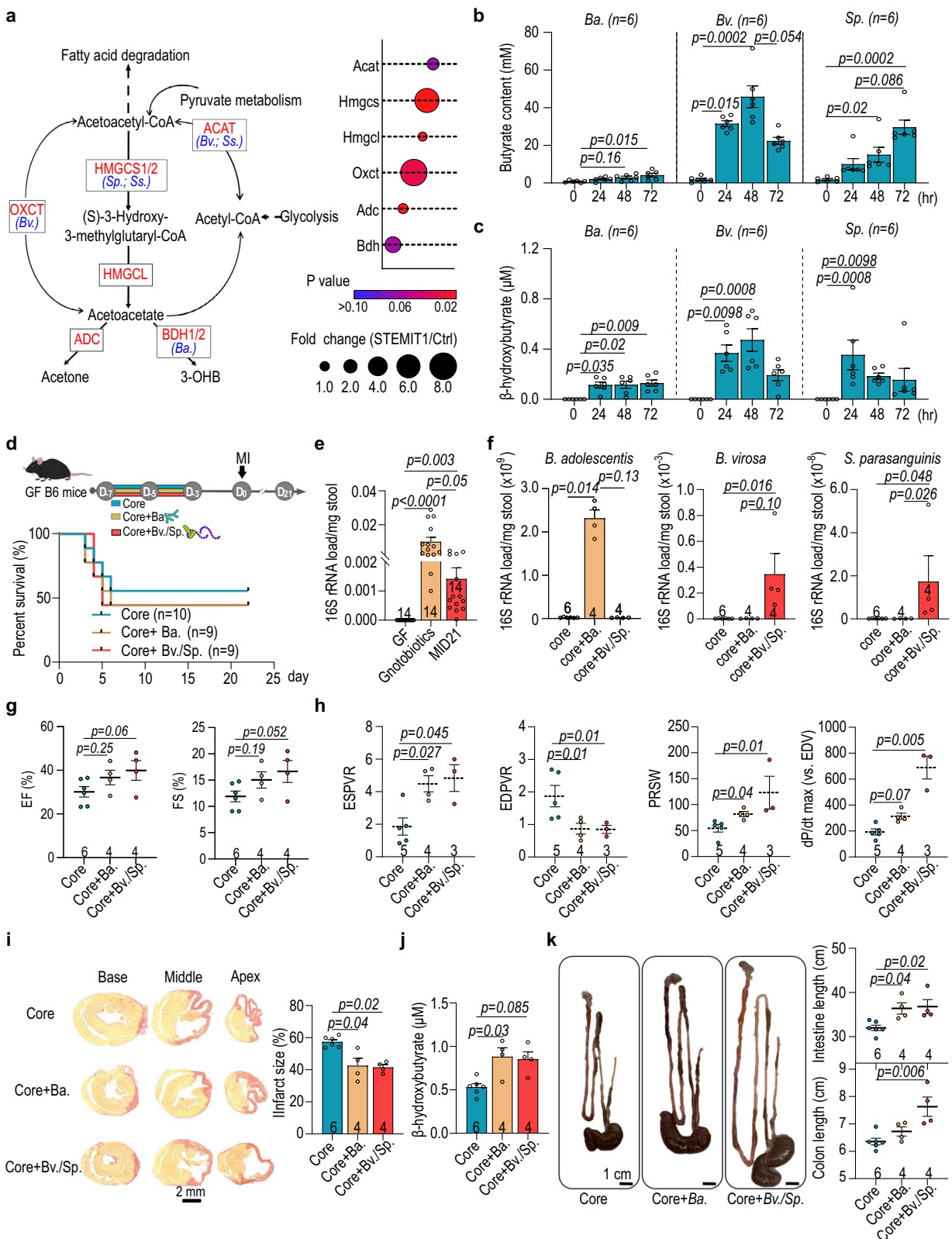

hydroxybutyrate observed in Fig. 5c. To directly test whether *B. ado-lescentis*, *B. virosa* and *S. parasanguinis* could influence post-injury cardiac function, we inoculated GF mice with a "core" bacterial community to serve as a control. This core community included eight species commonly found in human gut microbiota with limited butyrate-producing capabilities (*Anaerotruncus colihominis*, *Bacteroides caccae*, *Bacteroides thetaiotaomicron*, *Clostridium symbiosum*, *Collinsella aerofaciens*, *Coprococcus comes*, *Providencia stuartii* and *Ruminococcus torques*)[23]. Other animals received a combination of core with Ctrl-predominant *B. adolescentis* (Core+ *Ba*), or STEMI-associated *B. virosa/S. parasanguinis* (Core+ *Bv/Sp*) prior to MI surgery. Mice were then observed for twenty-one days after MI induction (Fig. 6d–f and Supplementary Fig. 10b). Post-MI survival rates were similar for mice from all three groups (Fig. 6d). We confirmed successful colonization

**Fig. 6 | Inoculation of *B. adolescentis* and *B. virosa* improves post-MI cardiac function in germ-free mice. a** Shotgun metagenomics analysis revealed the enrichment of ketogenesis in STEMIT1 samples. Abundance of genes encoding ketogenic enzymes in Ctrl and STEMI samples is presented in the dot plot. **b** Butyrate production capabilities of *B. adolescentis*, *B. virosa* and *S. parasanguinis*. **c** β-hydroxybutyrate production capabilities of *B. adolescentis*, *B. virosa* and *S. parasanguinis*. **d** Experimental design of the myocardial infarction (MI) model in gnotobiotic mice inoculated with *B. adolescentis*, *B. virosa* and *S. parasanguinis* (upper panel). The survival curve of gnotobioic mice subjected to MI for 21 days (lower panel). **e** Changes in bacterial load in gnotobiotic mice determined with 16S rRNA qPCR. **f** Bacterial load of *B. adolescentis*, *B. virosa* and *S. parasanguinis* in gnotobiotic mice. **g** Echocardiographic analysis of the left ventricular ejection fraction (EF, %) and fraction shortening (FS, %) in gnotobiotic mice on MI day 21. **h** Post-MI cardiac function analysis in gnotobiotic mice, evaluating ESPVR, EDPVR, PRSW and dP/dt max (vs. EDV) using cardiac catheterization. **i** Representative histology of cardiac infarct size (left panel) and statistics (right panel) in gnotobiotic mice on MI day 21. Cardiac tissues were stained with picrosirius red to label fibrosis. **j** Colorimetric analysis of the plasma level of β-hydroxybutyrate in gnotobiotic mice on MI day 21. **k** Representative images of the gut in gnotobiotic mice (left panel), intestinal length (upper right panel) and the length of colon (lower right panel) on MI day 21. The number of biologically independent samples and mice are indicated in each chart. Data were analyzed with Kruskal–Wallis test followed by FDR correction. Data are represented as mean ± SEM. ACAT Acetyl-CoA acetyltransferase, HMGCS 3-hydroxy-3-methylglutaryl-CoA synthase, HMGCL 3-hydroxy-3-methylglutaryl-coenzyme A (HMG-CoA) 2 lyase, OXCT succinyl-CoA:3-oxoacid CoA transferase (OXCT), ADC acetoacetate decarboxylase, BDH beta-hydroxybutyrate dehydrogenase, *Bifidobacterium adolescentis* (*B. adolescentis*); *Butyricimonas virosa* (*B. virosa*); *Streptococcus parasanguinis* (*S. parasanguinis*).

with *Ba* or *Bv/Sp* (Fig. 6f) and these mice showed improved cardiac function compared to mice receiving core bacteria only. Mice transplanted with *Ba* or *Bv/Sp*, showed positive trends in EF and FS (Fig. 6g and Supplementary Fig. 10c). Higher ESPVR, PRSW and dP/dt max (vs. EDV) as well as lower EDPVR were measured in *Ba* or *Bv/Sp* gnotobiotic mice, indicating better preservation of cardiac mechanical properties (Fig. 6h). Moreover, the average infarct size after twenty-one days was smaller in both *B. adolescentis*, and *B. virosa/S. parasanguinis* gnotobiotic mice (Fig. 6i), indicating more cardiomyocyte protection following injury. These mice also showed an increase in the plasma levels of β-hydroxybutyrate (Fig. 6j). We also found that *Bv/Sp* gnotobiotic mice displayed a longer intestine and colon than those inoculated with core bacteria only (Fig. 6k), suggesting the impact of butyrate-producers on intestinal homeostasis and morphology[27]. These data were highly suggestive that gut microbiome ketogenesis has a strong influence of MI outcome. However, endogenous hepatic ketogenesis could confound the results. To address this, we used *hmgcs2-Δexon2* (HMGCS2-deficient/ HMGCS2def) mice as a platform for inoculation with *Ba* or *Bv/Sp*. These mice are deficient in endogenous ketogenesis and lack β-hydroxybutyrate (Fig. 7a–d and Supplementary Fig. 11a); thus, any plasma β-hydroxybutyrate must be derived from the microbiota. These mice also have poor outcomes following MI. We confirmed that the presence of butyrate-producing bacteria in the gut increased plasma levels of β-hydroxybutyrate in HMGCS2def mice at MID14 (Fig. 7e). Additionally, the HMGCS2 protein levels were similar in HMGCS2def mice with/without the presence of butyrate-producing bacteria, both which were less than the HMGCS2 protein level in the control (Wt) mice (Supplementary Fig. 11b). Furthermore, HMGCS2def mice in the presence of *Ba*, *Bv/Sp* showed better cardiac function, with higher EF and FS, and smaller infarct sizes than the baseline HMGCS2def counterparts (Fig. 7f–h and Supplementary Fig. 12). Together, these data directly show a cardiac protective role of *B. adolescentis* and *B. virosa/S. parasanguinis* and demonstrate the contribution of bacteria-associated ketone body metabolism in post-MI cardiac protection.

## Discussion

MI induces a profound alteration of cardiac metabolism and affects other metabolically active tissues[28,29]. In the standard mouse MI model, we have previously shown that a post-MI decrease in *Lactobacillus* and SCFA production, where supplementation of *Lactobacillus* or SCFAs improved cardiac function[9]. However, our data showed that the mouse and human gut microbiota are dissimilar, with human *Lactobacillus* comprising only 0.25 ± 0.06 % of the identified bacteria, compared to 26.25 ± 5.61 % in mice. Therefore, we were motivated to carry out a similar study in humans. We focused on patients with confirmed STEMI, which is the most severe and life-threatening type of acute coronary syndrome[30]. Combining genomics and metabolomics approaches, we identified an increase in butyrate-producing bacteria (e.g. *Butyricimonas virosa* and *Streptococcus parasanguinis*) in STEMI

gut microbiomes, accompanied by the enrichment of ketone body metabolism at the acute phase (Figs. 1 and 4). These phenomena were conserved in the nonhuman primate model of IR, thus significantly strengthening the evidence that these butyrate-producers and ketone body metabolism may be biomarkers for STEMI. The nonhuman primate model has several advantages; being genetically, physiologically, biochemically and metabolically more similar to humans. This experiment also allowed us to minimize genetic, dietary and environmental factors which may have influenced our human-derived data, including bias of the machine learning model. Moreover, we were able to introduce consistent ischemic size and duration in the nonhuman primate hearts, and collect samples from the same individuals at precise time points, including prior to IR. Thus, the data obtained are expected to be more robust than those from the less-controlled human study and combined, they provide a demonstration of clinical relevance and strong scientific evidence.

During heart failure, cardiac metabolism shifts from fatty acid oxidation towards glycolysis and ketone body metabolism as the source of ATP generation[31–33]. Elevated level of circulating ketone bodies have been reported to negatively correlate with the STEMI outcome in a clinical study[24]. Further to this, studies have shown therapeutic effects of exogenous ketones given in MI rodent models; our data support these findings[25,26]. However, our study found that this effect was blunted in microbiome-depleted animals, which may be due to reduced β-hydroxybutyrate production (Figs. 5 and 6). This observation could be investigated further in the future studies.

Ketone bodies, produced mainly in the liver, can also be generated in the heart during fasting or following injury[28,34]. In the current study, we have confirmed the butyrate-producing ability of *Butyricimonas virosa* and *Streptococcus parasanguinis* in vitro (Fig. 6b). Surprisingly, we also observed the capability of *Bifidobactcerium adolescentis* and butyrate-producers (*Butyricimonas virosa* and *Streptococcus parasanguinis*) to generate ketone bodies; in both in vitro cultures and gnotobiotic experiments. These subsequently aided in post-MI cardiac function, and shaped the intestinal pathophysiology (Fig. 6c, j, k). Although the precise pathway is currently unknown, we speculate that these butyrate-producers may adopt similar pseudoketogenesis as the myocardium, in which butyrate undergoes ketogenesis to form acetoacetylCoA, then acetoacetate and β-hydroxybutyrate[35]. In mouse models, butyrate and β-hydroxybutyrate have been shown to modulate differentiation of Treg cells or expansion of Th17 cells to influence MI outcomes[36–39]. Furthermore, it is likely that butyrate-producers modulate intestinal homeostasis after MI through modulating the bowel inflammation. We found alteration of intestinal and colonic lengths as reported in the inflammatory bowel diseases, as well as higher circulating pro-inflammatory cytokines[22]. This raises an interesting possibility that bacterial ketone bodies may exert post-MI cardiac protection in part through modulating the intestinal immune system.

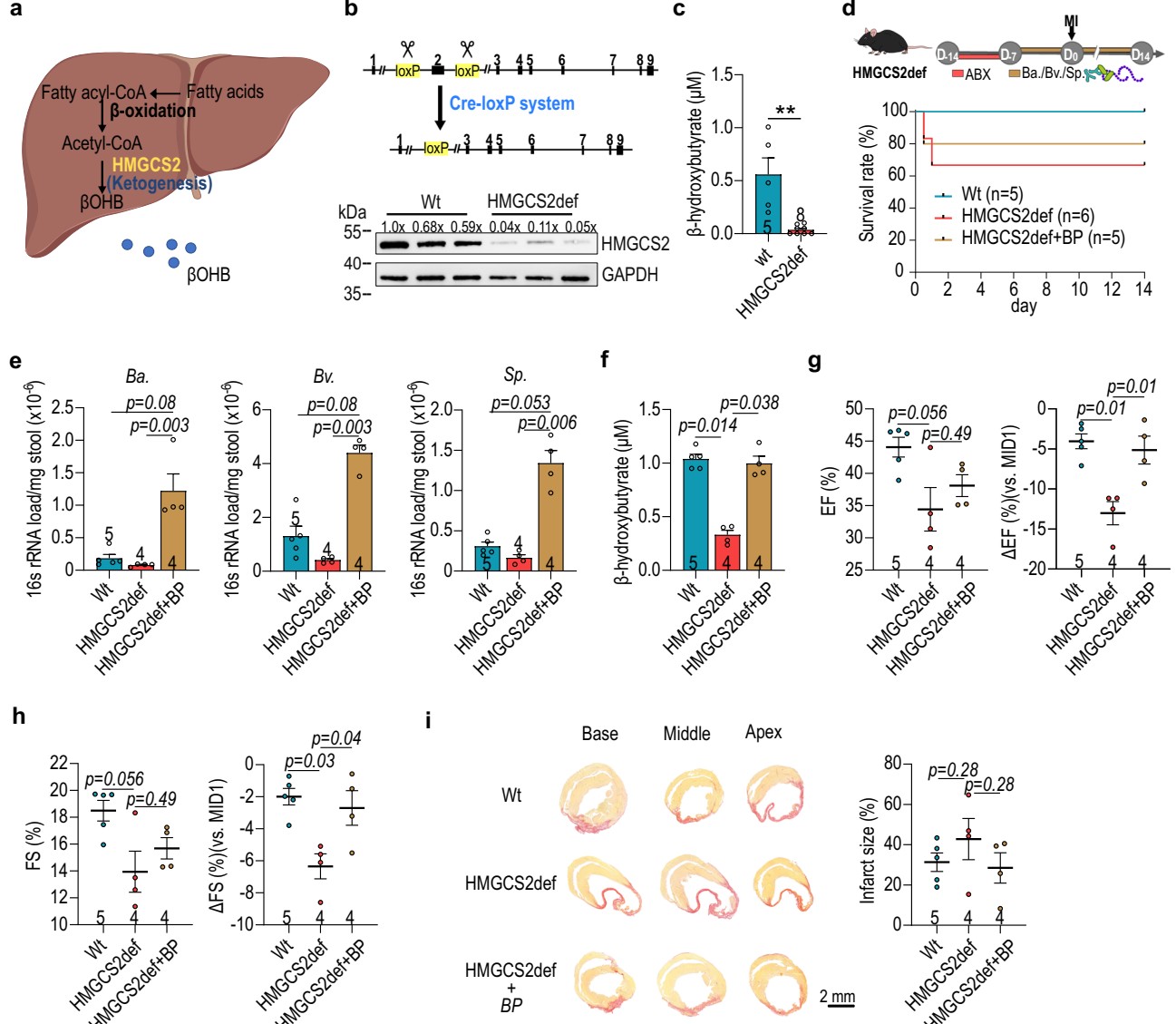

**Fig. 7 | Post-MI cardioprotection of butyrate-producers in *hmgcs2*-Δexon2 (HMGCS2-deficient/HMGCS2def) mice. a** Schematic illustration of liver ketogenesis. **b** Schematic illustration of HMGCS2def mice using Cre-LoxP system (upper panel) and the level of HMGCS2 protein in liver (lower panel). **c** The plasma level of β-hydroxybutyrate in HMGCS2def mice before cardiac injury with colorimetric assay (Wt, *n* = 5; HMGCS2def, *n* = 8). **d** The schematic illustration of experimental design for butyrate-producing (Bp) bacteria transplantation in 16-week-old female HMGCS2def mice (upper panel) and the post-MI survival rate of HMGCS2def mice (lower panel). **e** Loading of *Bifidobacterium adolescentis* (*B. adolescentis*),

*Butyricimonas virosa* (*B. virosa*) and *Streptococcus parasanguinis* (*S. parasanguinis*) in HMGCS2def stool. **f** The plasma level of β-hydroxybutyrate in HMGCS2def mice after MI determined with colorimetric assay. **g**–**h** Left ventricular ejection fraction (EF) (**g**) and fraction shortening (FS) (**h**) of HMGCS2def mice with/without butyrate-producer inoculation after MI. **i** Representative images of cardiac infarct size labeled with picrosirus red and statistics. The number of biologically independent mice are indicated in each chart. **e**–**i** Data were analyzed with the Kruskal–Wallis test followed by FDR correction. Data are represented as mean ± SEM.

Our study is not without limitations. The first is the relatively small human cohort size upon which we trained the machine learning model. To minimize variability we recruited participants from three medical centers over two years, accepting only patients with confirmed STEMI, and excluding all other types of MI. We also applied stringent inclusion/exclusion criteria, such as excluding previous CVD or MI, which significantly limited our patient recruitment ability. A homogeneous, small cohort size causes unavoidable overfitting, but our model suffered only a mild degree of overfitting, producing an AUC value of 0.88 for the test set (Supplementary Table 4) from a value of 1.0 for the training set. The performance of our model surpasses a previous report using microbiota database of 951 participants from the American Gut Project to predict CVDs[20]. We also lacked patient dietary records, and could not control for all variables,

including a higher proportion of hyperlipidemia in the STEMI group compared to the control group. While our machine learning model provided favorable diagnostic power in the cohort recruited, further validation using a broad external cohort will be needed to address other possible biases, such as diet, age, ethnicity, sex, and socio-economic level; all of which might affect the performance of the machine learning model, especially in a clinical setting. Another limitation of machine learning models is the "black box phenomenon", or incomprehensible internal predictive processes, which make it difficult to understand the true biological significance of results. To overcome many of these limitations, we applied the nonhuman primate disease model where we could control for diet and other variables, as well as collecting samples over more time points. There are also challenges associated with identifying bacteria-

derived β-hydroxybutyrate in vitro and in vivo. To enhance the validation of bacterial ketogenesis, it will be necessary to cultivate bacteria in a β-hydroxybutyrate-free medium and subsequently analyze the bacteria-conditioned medium via LC-MS/MS. Conducting gnotobiotic mouse experiments involving bacteria lacking key genes in ketogenesis could also provide further insights. Additionally, employing mice with whole body *hmgcs*2 knockout would help mitigate potential confounding factors, such as the contribution of residual HMGCS2 protein in the HMGCS2-deficient mice.

Our systemic approach, starting from an observation in human STEMI samples, validation in nonhuman primate IR, and direct experimental evidence in rodents, clearly identifies alteration of gut butyrate-producers after cardiac injury. Furthermore, these bacterial populations play a direct role in post-MI outcomes which may be attributable to ketogenesis. Taken together, our results provide mechanistic cause-and-effect evidence to support previous studies, and provide new insights into the gut-heart-axis during post-MI cardiac repair. These findings can likely be exploited for therapeutic purposes in the future.

## Methods

### Study cohort and sample collection

The study was conducted with patients admitted to National Cheng-Kung University Hospital (NCKUH; from January 2018 to April 2021) (south Taiwan), China Medical University Hospital (CMUH; from March 2019 to April 2021) (central Taiwan), and Far Eastern Memorial Hospital (FEMH; from May 2019 to April 2021) (north Taiwan), Taiwan. The ST-elevation myocardial infarction (STEMI) status was diagnosed with electrocardiography and catheterization by board-certified cardiologists at three medical centers in Taiwan. Pregnant female patients and patients with cancer, organ transplantation, previous myocardial infarction and decompensated heart failure in the past five years were excluded. Stool and plasma samples were collected after primary percutaneous coronary intervention (PCI) and fourteen to thirty days after intervention. The samples and clinical information collection in this study were with the approval of the institutional review board of participant institutes (IRB on Biomedical Science Research Academia Sinica, AS-IRB02-110151; Institutional Review Board NCKUH, 8800-4-03-005; Research Ethics Review Committee CMUH, CMUH108-REC3-016(CR-2); Research Ethics Review Committee FEMH, 107175-E). All participants have signed the informed consent forms. Patients did not restrain from eating nor did they have medication that induces ketogenesis during sample collection.

### Animals

C57Bl/6J specific pathogen free (SPF) and germ-free (GF) mice were provided by National Laboratory Animal Center (NLAC), Taiwan. The GF mice were kept in isolators. The HMGCS2 (3-Hydroxy-3-Methyl-glutaryl-CoA Synthase 2)-deficient (*hmgcs2*-Δexon2/HMGCS2def) mice were generated in house by removing the second exon of the nine exons of *hmgcs2* gene using the Cre-loxP system[40]. Mice were all under a 12-hour light/dark cycle with unlimited access to sterile food (chow diet; Cat No. 5053; LabDiet, USA) and water. All mice used in this study were 10-12 weeks old and male. The mouse experiments in this study have been approved by Academia Sinica Institutional Animal Care and Use Committee and NLAC Animal Care and Use Committee respectively (IACUC No. 18041211). The surgeon was blinded to experimental groups; mice of different experimental groups were given to the surgeon randomly. For nonhuman primate experiment, 10-year-old rhesus macaque (*Macaca mulatta*) monkeys were obtained from the Wisconsin National Primate Research Center and the experiments was approved by the experimental animal committee of UW-Madison (IACUC No. G006084-A07). All procedures of the nonhuman primate experiment were approved by the center.

### Nonhuman primate cardiac ischemia-reperfusion (IR) injury model

The 10-year-old macaques were anesthetized with intramuscular injection of ketamine and midazolam, intravascular propofol as needed, intubated and anesthetized using isoflurane to maintain anesthesia. In addition, animals were administered intravascular ketamine and fentanyl as continuous rate infusions. The heart was accessed through left-sided thoracotomy and the mid-left anterior descending (LAD) coronary artery were temporary ligated for 90 min.

### Stool DNA extraction

DNA was extracted from frozen fecal samples with the bead-beating method. The human and mouse stool DNA was extracted with Easy-Prep Stool Genomic DNA kit (Cat No. DPT-BC28; Biotools, Taiwan) and the rhesus macaque stool DNA was extracted with innuPREP Stool DNA Kit (Cat No. 845-KS-7010050, Analytik Jena GmbH, Germany) following manufacturers' instructions. After final precipitation, the DNA samples were resuspended in TE buffer and stored at −80 °C before further analysis.

### 16S rRNA sequencing

The V3-V4 region of 16S rRNA gene was amplified by a specific primer set (319F: 5′-CCTACGGGNGGCWGCAG-3′, 806R: 5′-GACTACHVGGG-TATCTAATCC−3′) following the 16S Metagenomic Sequencing Library Preparation procedure (Illumina, UAS). 12.5 ng of genomic DNA was used for the PCR reaction with KAPA HiFi HotStart ReadyMix (Cat No. KR0370−v14.22; Roche, Switzerland) under the PCR condition: 95 °C for 3 min; 25 cycles of: 95 °C for 30 s, 55 °C for 30 s, 72 °C for 30 s; 72 °C for 5 min and hold at 4 °C. The PCR products were monitored on a 1.5% agarose gel and those with a bright main strip of 500 base pairs were purified using AMPure XP beads (Cat No. A63882; Beckman Coulter, USA). To prepare the sequencing library, the dual indices and Illumina sequencing adapters were added to the 16S rRNA V3-V4 PCR amplicons using Nextera XT Index Kit (Cat No. FC-131-1096; Illumina, USA). Quality of the indexed PCR products was assessed on the Qubit 4.0 Fluorometer (ThermoFisher Scientific, USA) and Qsep100TM system. Equal amount of the indexed PCR products was mixed and sequenced on an Illumina MiSeq platform (Illumina, UAS) to generate paired 300-base pair reads. The sequence results were processed using QIIME 2 (version 2020.11)[41]. Primer-trimmed sequences were clustered to amplicon sequence variants (ASVs) using the q2-dada2 plugin and sequences with anonymous bases and chimera were filtered. The phylogenetic information was obtained by classifying each representative sequence with a pre-trained Naive Bayes classifier (Silva database v.138) and the qiime-feature-classifier function. Analysis of taxonomic diversity and LEfSe (Linear discriminant analysis (LDA) Effect Size) were performed with Galaxy platform and MicrobiomeAnalyst[42,43]. Samples were further selected for whole-genome shotgun sequencing based on the algorithm of Microbiomes: Picking Interesting Taxa for Analysis (microPITA), falling into the unsupervised criteria of diverse, features and representative but not in the criteria of extreme[44].

### Whole-genome shotgun sequencing

Whole-genome shotgun sequencing of human fecal samples was carried out on the NovaSeq 6000 platform (Illumina, USA) and paired-end sequenced to a targeted data size of 6.0 Gb. The original data were transformed into raw sequenced reads by CASAVA base calling and stored in FASTQ format. After removal of adaptor sequences and poor-quality bases, the data were processed on Bowtie2 (v2.3.5.1) to remove contaminating host sequences with the human reference genome GRCh38 as the database[45]. MEGAHIT (v1.1.3) with settings '- -k-list 21,29,39,59,79,99,119,141' was applied for de novo assembly of the filtered reads[46]. CD-HIT (v4.7) with 95% identity was used to construct a nonredundant gene catalog, and the clean reads from each sample

were mapped to the initial gene catalog via BWA (v0.7.17-r1188)[47]. The assembled Unigenes were blasted with the NCBI Refseq using DIAMOND software (0.9.22) for identification of bacterial, fungi, archaea and viruses[48,49]. The taxonomical level of each gene was determined using the lowest common ancestor (LCA) algorithm. Gene clusters were functionally annotated by comparison with the Kyoto Encyclopedia of Genes and Genomes (KEGG) and protein database (GhostKOALA)[50].

## Fecal microbiota transplant (FMT)

The human fecal samples were dissociated in sterile PBS at the concentration of 100 mg/ml and the solution were passed through a 100-μm cell strainer (Falcon, USA). The human fecal microbiota transplant was given to the GF mice with 300 μl/time every other day via oral gavage for one week as previously described[21]. Four independent donors (two Ctrl donors and two STEMI donors) were used and each recipient group was housed in a separated isocage to avoid contamination. For SPF mice, the mice were treated for one week with an antibiotic cocktail containing 0.25 mg/ml ampicillin (Cat No. SI-A9518-25G; Sigma-Aldrich, USA), 0.25 mg/ml metronidazole (Cat No. M1547-25G; Sigma-Aldrich, USA), 0.25 mg/ml neomycin (Cat No. N1876-25G; Sigma-Aldrich, USA) and 0.125 mg/ml vancomycin (Cat No. V2002-5G; Sigma-Aldrich, USA). The mice were then colonized with 300 μl/time of the human microbiota transplant for three time every other day prior to surgery.

## Butyrate supplementation experiment

The male 12-week-old C57Bl/6J mice were treated with 1 M of sodium butyrate (Cat No. 303410; Sigma-Aldrich, USA) one day after LAD ligation for twenty-one days. The influence of the gut microbiota were tested by co-treatment of a combination of antibiotics (ABX; 0.25 mg/ml ampicillin, 0.25 mg/ml metronidazole, 0.25 mg/ml neomycin and 0.125 mg/ml vancomycin) (all from Sigma-Aldrich, USA). For dose-dependent experiment, the mice were treated with 0.03 M, 0.3 M and 1 M of sodium butyrate for twenty-one days after surgery. 1 M is the saturation concentration of butyrate in double-distilled water.

## Bacteria preparation

The bacteria used for the gnotobiotic experiments were purchased either from the Bioresource Collection or Research Centre (BCRC) Taiwan or Microbe Division of RIKEN BioResource Research Center (JCM, RIKEN BRC) Japan as indicated in the parentheses after each bacterium. The core community for gnotobiotic experiment included *Anaerotruncus colihominis* (JCM, 15631), *Bacteroides caccae* (JCM, 9498), *Bacteroides thetaiotaomicron* (BCRC, 08B0358), *Clostridium symbiosum* (BCRC, 14487), *Collinsella aerofaciens* (BCRC, 16136), *Coprococcus comes* (JCM, 31264), *Providencia stuartii* (BCRC, 13998) and *Ruminococcus torques* (JCM, 6553). Experimental bacteria included *Bifidobacterium adolescentis* (BCRC, 14606), *Butyricimonas virosa* (JCM, 15179[T]) and *Streptococcus parasanguinis* (BCRC, 14739). Aerobic *P. stuarii* and *S. parasanguinis* were expanded in Todd Hewitt broth (Cat No. T1438; Sigma-Aldrich, USA) at 37 °C. *A. colihominis*, *B. caccae*, *B. thetaiotaomicron*, *B. virosa*, *C. aerofaciens*, *C. comes* and *C. symbiosum* were expanded in GAM broth (Cat No. M1801; HIMEDIA LABORATORIES, India) under anaerobic condition at 37 °C. *B. adolescentis* was cultured in Lactobacilli MRS broth (Cat No. 288130; Becton Dickinson, USA) under anaerobic condition at 37 °C.

## Gnotobiotics experiment

GF C57BL/6 mice were maintained in a controlled environment in gnotobiotic isocages (TECNIPLAST S.p.A., Italy) under a 12-h light/dark cycle with sterile food and water. The 12-week-old GF male mice were inoculated trice every other day by oral gavage with 0.3 ml of mixed bacteria culture ($10^9$ CFU for each bacterium), 'core' community, 'core plus *Bifidobacterium adolescentis*', and 'core plus *Butyricimonas virosa* and *Streptococcus parasanguinis*' in gnotobiotic isocages. The gnotobiotic mice were subjected to myocardial infarction surgery after bacteria colonization. For HMGCS2-deficient mice experiments, 14-week-old female mice were treated with a combination of ABX for fourteen days prior to bacteria inoculation. Started from seven days prior to MI surgery, the HMGCS2-deficient mice were supplementation of *Bifidobacterium adolescentis*, *Butyricimonas virosa* and *Streptococcus parasanguinis* ($10^9$ CFU for each bacterium) daily in 5% skim milk and continued throughout the whole observation.

## Multiplex Immunoassay Profiling

Systemic cytokine levels were determined with the ProcartaPlex™ Multiplex Immunoassay (Invitrogen, USA) following manufacturer's instruction. Mouse plasma samples were analyzed with the Plex mouse ProcartaPlex panel (EPX170-26087-901, Invitrogen) and signals were detected on the Luminex 100/200 system.

## LC-MS untargeted metabolic profiling

The human and monkey plasma samples were deprived of protein by mixing 10 μl of the plasma samples with 90 μl of methanol. The mixture was centrifuged at $15,000 \times g$ for 15 min and supernatant was filtered through a 0.22 μm PP membrane (RC-4, Sartorius, Göttingen, Germany). Three repeated analysis was performed for each sample using Agilent 1290 UHPLC system coupled with 6540-QTOF (UHPLC-QTOF) (Agilent Technologies, Santa Clara, CA). Acquired total ion chromatogram was processed with True Ion Pick (TIPick) algorithm for background subtraction and peak picking. Peaks were identified by matching m/z to an established in-house database: the National Taiwan University MetaCore Metabolomics Chemical Standard Library. Obtained potential metabolite features were normalized by sum of total peak area, log transformed, and autoscaled (mean-centered and divided by the standard deviation of each variable) prior to statistical analysis using MetaboAnalyst5.0[51].

## $^1$H-NMR metabolite profiling

The centrifugal filters (Amicon Ultra 0.5, MWCO3KDa; Merck, Germany) were washed trice with deionization and distilled water at 13,800×g for 20 min. 300 μl of the human plasma samples were filtrated through the centrifugal filter at 13,800×g for 90 min. The filtrates were mixed with 100 μl of phosphate buffer (77.4 mM $N_aH_2PO_4$, Cat No. S5011-500G; 22.6 mM $Na_2HPO_4$, Cat No. 255793-10G; Sigma-Aldrich, USA) in $D_2O$ (Cat No. AL-151882-100G; Sigma-Aldrich, USA) containing 100 μM 3-(Trimethylsilyl)propionic-2,2,3,3-d$_4$ acid (TSP; Cat No. 269913-1G; Sigma-Aldrich, USA). The solution was made up to 600 μl with phosphate buffer in $D_2O$ and transferred to 5 mm NMR tubes (OPTIMA, Japan) for NMR analysis. One-dimensional $^1$H-NMR spectra with water pre-saturation were measured for all samples at 298 K on a BRUKER AVANCE III 600 MHz spectrometer equipped with a TXI ($^1$H/$^{13}$C/$^{15}$N) 5 mm CryoProbe (Bruker, USA). Each spectrum was acquired with 128 scans, using a recycle delay (d1) of 2 s, and was processed with Topspin 2.1 (Bruker, USA). Metabolites annotation and quantification were processed using Chenomx NMR suite 8.5 (Chenomx Inc., Canada).

## HPLC measurement of butyrate

Fecal samples collected immediately after animal sacrifice were kept at −80 °C before analysis. To measure the fecal butyrate level, 100 mg of fecal samples were homogenized in 1 ml of HPLC grade water and centrifuged at 14,000×g to remove bacterial cells and debris. 20 mM of butyrate was added to the supernatant as a reference. The level of butyrate was analyzed using a Waters Alliance e2659 HPLC (Waters, USA) with a Waters 2489 UV/Vis detector (Waters, USA) and a Waters XBridge C18 RP column (5 μm, 4.6 × 250 mm) (Waters, USA). 0.01 M of $H_2SO_4$ was used as the mobile phase. Butyrate was identified by

comparing sample peak retention time with standards and the concentration was determined using Waters Empower 3 (Waters, USA).

## Colorimetric analysis of plasma β-hydroxybutyrate

Plasma β-hydroxybutyrate level was determined with β-Hydroxybutyrate Colorimetric Assay Kit (Cat No. ab83390; Abcam, UK) following manufacturer's instruction. 5 μl of plasma was diluted in 45 μl of assay buffer. The diluted samples were mixed well with the reaction mix (46 μl of assay buffer, 2 μl of enzyme mix and 2 μl of substrate mix) and protected from light at room temperature for 30 min. The mixture was measured at OD 450 nm. For bacterial culture, the plain culture medium was also tested to account for background signal.

## Bacterial β-hydroxybutyrate quantification

Production of β-hydroxybutyrate by butyrate-producers were determined using LC-MS. 100 μl of bacterial culture medium was first mixed with 40 ng of rac 3-Hydroxybutyric Acid-d4 Sodium Salt (Cat No. TRC H833022-1MG; TRC, Canada) and then treated with 300 μl of acetonitrile (Cat No. JT-9017-88; Fisher Scientific, USA) and centrifuged at 14,000 × $g$ at 4 °C for 5 min to remove protein. The 10-fold diluted supernatant was injected to the QTRAP® 5500 LC-MS/MS System (SCIEX, USA) to detect the presence of β-hydroxybutyrate. We used ACQUITY UPLC™ BEH C18 column (1.7 μm, 2.1 mm × 50 mm) (Waters, USA) to separate the samples. 0.1% of ammonia solution (Cat No. 1054280500; Sigma-Aldrich, USA) and 0.1% of formic acid (Cat No. 27001-500ML-R; Sigma-Aldrich, USA) in acetonitrile were used as the mobile phase. Samples were ionized using the negative ion electrospray ionization mode. Peaks were identified by matching m/z to an established in-house standard of purified sodium 3-hydroxybutyrate (Cat No. H3145; Sigma-Aldrich, USA).

## Histology examination

Paraffin-embedded tissue sections were dewaxed, rehydrated and subsequently stained with appropriate staining protocols. For determination of the infarct size, the heart sections were stained with picrosirius red for 1 h. Slides were washed trice with tap water. The stained sections were dehydrated, cleared in xylene and mounted in a resinous medium (SUB-X-MOUNTING-MEDIUM; Leica Surgipath, USA). The size of cardiac myocytes was determined by staining the tissue sections with WGA-488 (Cat No. W11261; Invitrogen, USA) and anti-α-actinin antibody (Cat No. A7811; Sigma-Aldrich, USA) and mounted in fluorescent mounting medium (Cat No. F4680; Invitrogen, USA). Anti-mouse IgG antibody conjugated with Alexa 568 (Cat No. A-11004; Invitrogen, USA) was used to recognize the signal of anti-α-actinin antibody. For intestine pathophysiology, rehydrated gut sections were first stained with haematoxylin (Cat No.HMM3800; ScyTek Laboratories, USA) for 2 min and rinsed under running tap water. The sections were then stained with eosin (Cat No. E6003-25G; Sigma-Aldrich, USA) for 2 min and rinsed under running tap water. The stained gut sections were dehydrated, cleared and mounted in a resinous medium. Images were captured either by the LSM700 confocal microscope (Carl Zeiss, Germany) or Pannoramic 250 FLASH II (3DHISTECH, Hungary), and quantified by ImageJ software. The brightness and contrast adjustments of immunofluorescence images were applied equally to all images within a series to improve visual clarity.

## Supervised machine learning

The study population consisted of 147 stool 16S rRNA V3-V4 NGS data, including 70 controls and 77 STEMI patients. The NGS data were processed with QIIME 2 (version 2020.11) to generate amplicon sequence variants (ASVs). Raw or normalized ASV abundance was utilized to construct the database for supervised machine learning using the PyCaret package[52]. To precisely evaluate the prediction power of the NGS data, we computed ten 10-fold cross-validations which involved several strategies including splitting data, data normalization, and model optimization. In the splitting data, 70% of the samples were used randomly as the training set to build prediction model, and the remaining 30% samples were assigned as the test set to evaluate model performance. In the data normalization, ASV abundance was normalized with ratio (scaled each sample in the range of 0–1), Z-score (calculated the standard z-score for each feature), min-max (scaled and translated each feature in the range of 0–1), maxabs (scaled each feature such that the maximal absolute value of each feature will be 1.0) and robust (scaled and translated each feature according to the interquartile range). In the model optimization, sixteen modern machine learning algorithms were used to construct initial models. The top three initial models, according to their accuracies, were selected, fine-tuned, blended, and stacked. Among these optimized models the best model was chosen by accuracy to do calibration and to report the performance metrics of the test set. All parameters were defaults and all hyperparameters were automatically tuned by the PyCaret. The data, the source code, and the computation environment settings can be found in supplemental materials.

## Statistics & reproducibility

Statistical analysis and graph generation were performed with GraphPad Prism 8 (GraphPad Software, Inc., La Jolla, CA, USA). Data are presented as mean ± SEM. Statistical tests are described in the figure legends. Kruskal–Wallis test was applied for group analysis. Two-sided Student's t-test was used to analyze two independent groups. Survival rate was measured with Kaplan–Meier method and compared using Mantel–Cox log rank tests. The sample size was determined based on previous studies within the lab using these techniques with alpha equals to 0.05, beta equals to 0.2[9,40,53]. The relevant raw data were available in the Source Data.

## Reporting summary

Further information on research design is available in the Nature Portfolio Reporting Summary linked to this article.

## Data availability

The raw data generated in this study are provided in the Source Data file. The LC-MS data used in this study are available in the MetaboLights database under accession code MTBLS8802. The raw sequences used in this study are available in the NCBI SRA database [Accession number: PRJNA1031545]. Source data are provided with this paper.

## Code availability

The code for machine learning is provided in the supplementary Data 2 and is also available in the Zenodo database [https://doi.org/10.5281/zenodo.10032793].

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

## Acknowledgements

The authors would like to thank the Common Equipment Core and Pathology Core of Institute of Biomedical Sciences, and High Field Nuclear Magnetic Resonance Center at Academia Sinica for their technical support. We are grateful to Taiwan Biobank for providing the control plasma samples. We thank MetaCore at National Taiwan University for their support in LC-MS analysis. We also thank the National Laboratory Animal Center (NARLabs) for technical support in service of isolators. This work was supported by Taiwan Ministry of Science and Technology (MOST 110-2320-B-001-023-MY3; 111-2320-B-001-027-MY3; 111-2321-B-001-012; 111-2740-B-001-003 to P.C.H.H.), Taiwan National Health Research Institute (EX111-10907SI to P.C.H.H.) and Healthy Longevity Grand Challenge of US National Academy of Medicine and Academia Sinica (AS-HLGC-109-05 to P.C.H.H.) and The Translational Medical Research Program (AS-KPQ-109-BioMed to P.C.H.H.).

## Author contributions

P.C.H.H. supervise the research and acquired fundings; H.C.C. designed and directed the research, performed experiments, carried out data analysis, image analysis, wrote the manuscript and created the human silhouette, monkey and mouse images, using Microsoft PowerPoint; Y.W.L., K.C.C., Y.W.W., K.W.C., S.S.C. provided clinical samples; M.L.H., Y.C.C., C.M.C.C., R.P., A.W.M. and J.C. provided NHP samples; H.C.C., Y.K.C., M.Y.Y., C.J.L., R.P.P., P.J.L., S.C.R. and D.H.K.C. acquired data and specimen; H.Y.C and Y.Y.C. provided HMGCS2 engineered mice; Y.M.C. and E.S.C.S. wrote and verified the code for machine learning; H.C.C., Y.K.C. performed NGS analysis; P.C.H.H., D.J.L., J.R.L., D.C.W., M.J.H., T.A.H., C.L.E.Y., F.E.R. and T.J.K. reviewed and edited the manuscript.

## Competing interests

The authors declare no competing interests.
