## [Peer Review File · Nature Communications]

REVIEWERS' COMMENTS

Reviewer #1 (Remarks to the Author):

Chen et al. have done an overall thorough job addressing my original concerns.

A lingering major concern is that I am still not convinced about the degree to which bacteria produce beta-hydroxybutyrate (BHB). The UniProt accessions mentioned in the response are “inferred from homology” and given the weakest possible annotation score of 1/5. The colorimetric data shown in the response letter clearly shows BHB in the media controls, indicating that the assay is either cross-reacting with other compounds or the media itself contains BHB, complicating the interpretation of these data. There still is no MS2 data shown, making the mass spectrometry data preliminary.

I realize that it may not be possible to address these issues given the scope of the paper. At a minimum, additional qualifying statements should be added to the results and a more explicit text mentioning these limitations should be added to the discussion.

Furthermore, the response to my question about the HMGCS2 “KO” raised concerns. It appears that this is just a KO of exon 2 and that there is still protein produced. As such, it is more accurate to refer to this model as a knock-down and the residual BHB could still be derived from host. The authors should clarify if the residual protein is active and relabel the mouse HMGCS2 delta-exon2 or something to that effect. Together with the methodological concerns around BHB production by bacteria, the authors should temper their interpretation to acknowledge that some of what they are describing could be due to host metabolism.